# Liturgical Gratitude to God

Joshua Cockayne [1,*] and Gideon Salter [2,*]

1　School of Divinity, University of St Andrews, St Andrews KY16 9JU, UK
2　Department of Psychology, University of Sheffield, Sheffield S10 2TN, UK
*　Correspondence: jlc22@st-andrews.ac.uk (J.C.); g.salter@sheffield.ac.uk (G.S.)

**Abstract:** Gratitude to God is a core component of Christian liturgy; along with the countless hymns which express attitudes of thanks to God, Christian liturgy often includes acts of spoken gratitude, as well as prayers of thanksgiving. We argue that two aspects of liturgical gratitude distinguish it from gratitude more generally. First, liturgical gratitude is always scripted, leading to the worry that those who express gratitude do so disingenuously. Secondly, liturgical gratitude is always social in some way, as our gratitude is drawn into the worship of the community of the Church. The paper provides an account of liturgical gratitude that explores these two key distinctive features.

**Keywords:** gratitude; liturgy; group action; joint action; social ontology; gratitude to God

## 1. Introduction

One of the central features of Christian liturgy is the expression of gratitude. Gratitude might be expressed in spoken liturgy: for example, in reciting the words of a Psalm: "I will extol the LORD at all times; his praise will always be on my lips". (Psalm 34:1). Alternatively, gratitude might be expressed in the bodily movements of eating and drinking in the Eucharist (Eucharist literally means *thanksgiving*). Or we might express gratitude to God in song, in singing one of the thousands of hymns of gratitude which have been written, whether accompanied by rock guitars, or choir and organ. Along with confession and petition, gratitude is arguably one of the key attitudes expressed in Christian liturgy.

Much work has been done in recent years demonstrating the positive therapeutic effects of gratitude. There is evidence that gratitude improves one's mental health and emotional wellbeing in a variety of ways, from a greater sense of purpose and success in achieving personal goals to an improved physical and psychological health (Emmons 2013; Emmons and Stern 2013; Petrocchi and Couyoumdjian 2016).

Given the prominence of gratitude in Christian liturgies, together with the positive therapeutic effects of gratitude, it seems reasonable to assume that Christian liturgy is good for us (even if gratitude may not be the only (or even primary) reason for this effect). Taking time to consider the good gifts we have received from God in song, in spoken liturgy, or in moments of quiet reflection provides an opportunity to grow in this important disposition of gratitude. This paper seeks to provide an account of liturgical gratitude.

In providing account of liturgical gratitude we outline two ways in which liturgical gratitude differs from gratitude more generally. First, liturgical gratitude is always scripted. Even in church traditions in which little formal liturgical script is present, there are always prescribed actions in the context of gathered worship. This need not mean that all liturgy includes a written script; if some kind of action is prescribed by the liturgy (e.g., you are expected stand up to sing, sit down to listen to sermons, etc.), then liturgy is scripted. This leads to the worry that liturgical gratitude might not really be gratitude—for if we are merely following a script, it might appear that we do not really mean it.[1] As many in the psychological literature on gratitude are keen to point out: feeling grateful is an important factor in being able to distinguish real gratitude from insincere gratitude, or worse. In liturgical contexts, many of us express gratitude to God, despite not always

feeling it.[2] To show why liturgical gratitude should count as gratitude even though it is scripted, we consider a recent discussion of liturgical action in the work of Terence Cuneo, which we think can explain how participating in acts of scripted worship can provide an opportunity for individuals to be grateful to God. Notably, for Cuneo, liturgical gratitude does not always involve *feeling* grateful. Building on Cuneo's account, we borrow from Everett Worthington's discussion of forgiveness to argue that it is important to distinguish the emotional component of gratitude with the decision to be grateful. What makes the decision to be grateful in liturgy valuable is not the emotions of the participant, but the fittingness of the actions in response to God.

Secondly, liturgical gratitude is always social in some way. Christian theology stresses that the worship of the Church, even in its seemingly most individual contexts must be understood as part of the community of the Church. However, while the literature on gratitude can tell us much about the positive effects of being grateful, there is very little work that focuses on gratitude as it occurs in groups. As Jo-Ann Tsang (2021) has recently observed, "the majority of research on gratitude focuses on single recipients of gratitude ... The prototypical situation is one benefactor providing a benefit to one recipient". (Tsang 2021, p. 27). However, as Tsang continues: "humans are social creatures. We experience benefits on a group level, as well". Leading her to ask, "Do people still experience gratitude, even if the intended recipient is broader than themselves?" (Tsang 2021, p. 27). To understand what is going on in acts of liturgical gratitude we need to reflect on the group dynamics present. Building on our proposed taxonomy of group gratitude (see Cockayne and Salter, forthcoming), we note the ways in which group liturgical thanks (e.g., "we thank you God for ... ") might express the gratitude of different subjects (e.g., the individual, the congregation, the Church). In doing so, we seek to offer an account of liturgical gratitude which is sensitive to its group dynamic.

Lastly, drawing these discussions to a close, we consider how liturgical gratitude may provide psychological benefits that are distinct from other forms of gratitude. Building on recent psychological literature on social bonding, we hypothesize that group gratitude may even have benefits beyond that of individual gratitude.

## 2. Liturgical Gratitude Is Scripted

One notable feature of liturgical gratitude that distinguishes it from acts of gratitude more generally is the presence of a script. For the presence of a script is foundational to what liturgy *is* (at least in the context of Christian worship).[3] Nicholas Wolterstorff, one of the leading figures in the study of Christian liturgy writes that,

> An enactment of a liturgy consists of the participants together performing scripted verbal, gestural and auditory actions, the prescribed purpose of their doing so being both to engage God directly in acts of learning and acknowledging the excellence of who God is and what God has done, and to be engaged by God. And the liturgy itself is that type of sequence of act-types that is enacted when the participants do what the script prescribes. (Wolterstorff 2018, pp. 29–30)

For Wolterstorff, participating in liturgy always requires following a script of some kind. However, this need not mean that liturgy is only found in formal or traditional contexts. In formal or traditional contexts, liturgical scripts are often written down in prayerbooks or service sheets, indicating when those present are to listen, to speak, to eat and drink, to sit and stand, and so on. In more informal contexts, there is still a script, even if this is implicit and unformalized; if no script were present, we would not know when to stand up and sit down, what words to sing and when, and to respond to the end of a prayer by saying "Amen". All liturgy has a certain set of prescribed actions, whether these are specified on paper or by social convention.

The scriptedness of liturgical gratitude raises the question of whether these are really instances of gratitude at all, or at the very least it should make us see that liturgical gratitude is unlike many other everyday instances of gratitude. As Wolterstorff notes "When one follows a liturgical script one does not choose one's own words; the words are prescribed.

Nor does one choose what saying or singing the words is to count as. The script, along with the linguistic conventions in force, determine the . . . significance of one's words". (Wolterstorff 2018, pp. 31–32). It seems possible that many who engage in liturgical thanks may do so with very little awareness of what they are gratitude for or to whom. Congregants may even participate in liturgical gratitude resentfully, feeling very little gratitude themselves. Moreover, we know that many go along to church and participate in liturgical gratitude without really ever *feeling* thankful to God. Think of the skeptical teenagers cajoled to church each week by their over-zealous parents, who reluctantly read along but have very little interest in doing so. Or consider the habitual churchgoer who has internalized the liturgy to the extent that they are more likely to be thinking about the previous day's football scores than attending to what God has given to them.

It is for precisely these reasons that some psychologists wish to emphasize the affective dimension of gratitude. Consider Robert Emmons's claim that, "Gratitude is an emotion, the core of which is pleasant feelings about the benefit received". (Emmons 2008, p. 469). It may be the case that one goes to church each week and expresses gratitude to God but rarely feels emotionally moved by the acts one is performing. Indeed, some even think that it is a feature of gratitude to God that an affective dimension is sometimes lacking. For example, Kent Dunnington suggests that if we

> focus on what is supposedly the essence of gratitude—grateful feelings—I suspect many Christians will agree that they fail to live up to the calling to be grateful to God. For many Christians, feelings of gratitude to God fall short of what one would expect given what Christians allege to be God's extraordinary beneficence. (Dunnington, forthcoming, p. 2)

On Emmons' characterization, these instances of liturgical gratitude which lack emotion would not count as genuine acts of gratitude, since they lacked an essential component, namely, an emotional response to God. Something seems intuitively right about this; for the mere presence of the word "thank you" seems insufficient for counting something as an expression of gratitude. So, an account of liturgical gratitude needs to explain what distinguishes proper gratitude from those who are merely reading a script.

In his book *Ritualized Faith*, Terence Cuneo provides an account of what it is to express gratitude through liturgy which can help us to make sense of the scripted nature of liturgical gratitude. First, Cuneo thinks, we need to make a distinction between liturgical acts (e.g., speaking, eating, prostrating) and the acts prescribed by these liturgical acts (e.g., blessing, petitioning, and thanking) (Cuneo 2016, pp. 156–57). Rather than maintaining that we must distinguish between gratitude and grateful liturgical actions in this context, Cuneo argues that,

> Actions of these latter sorts [i.e., speaking, eating, prostrating] do not merely accompany the linguistic acts prescribed by the liturgical script [i.e., blessing, petitioning, thanking], as if their function were merely to add emphasis to these linguistic acts. Rather, in the context of the liturgy, the kissing, prostrating, and eating also *count as* cases of engaging God by blessing, petitioning, and thanking God. In fact, these bodily actions are vivid cases of act-types by which a person can simultaneously perform multiple actions with expressive import without saying a thing. (Cuneo 2016, pp. 156–57; emphasis added)

Thus, as Cuneo goes on to argue, an act of liturgical gratitude, whether this involves speaking, eating, or kissing, does not depend on having a certain mental state whilst performing the action itself. Instead, he thinks, "Thanking, when all goes well, expresses gratitude. But to thank someone at some time, one needn't be feeling gratitude at that time". (Cuneo 2016, p. 157). In other words, the very act of eating or drinking bread and wine can count as an act of liturgical thanks when *all goes well.* Of course, eating and drinking may not be the only or even the primary mode of gratitude involved in the eucharist; the eucharistic prayer itself is an example of a scripted prayer of thanksgiving. What the acts

of consuming the elements help us to explore is those actions which do not obviously look like instances of gratitude.

So, what does Cuneo mean by "when all goes well?" Here, Cuneo refers to the notion of "aptness" to explain why some actions should count as gratitude and others not. He notes that an act of gratitude might be apt because it is accompanied by a relevant affective state, as in the following example:

> Suppose I write you a note thanking you for a gift that you have given me. If the writing of this note is accompanied by feelings of gratitude toward you, the expressive content of my action perfectly fits the mental state I am in when I write the note. As such, the performance of my action is especially apt. (Cuneo 2016, p. 157)

Similarly, the wrong kind of affective states may make an act of gratitude inapt, if, for example, "my action's expressive content fails to fit the mental state I am in when I write the note. It is thus an especially inapt or defective case of thanking". (Cuneo 2016, p. 157). However, for Cuneo, affective states are not the only measure by which we can assess the aptness of an expression of gratitude. If I write the note absent-mindedly, or because I am merely going through the motions (think of a manager writing thank you cards to hundreds of employees each Christmas), these can still count as acts of expressing thanks, according to Cuneo. He writes that,

> I may often fail to feel gratitude . . . their [i.e., the target of gratitude] actions may seem so remote in time that they fail to engage me emotionally. Still, arguably, my actions of thanking are highly apt. They are not apt because their expressive content fits the mental state I am in when I write these notes. Rather, they are apt because they are appropriate responses to what you have done on my family's behalf, which flows from a state of being resolved to express my family's gratitude. (Cuneo 2016, pp. 157–58)

According to Cuneo, the aptness of gratitude actions (whether eating/drinking or responding to a Eucharistic prayer) has to do with whether the responses are *appropriate*. While negative emotions may make certain actions defective or inapt, a lack of positive emotion is not enough to disregard certain actions as genuinely expressing gratitude. Thus, in the context of liturgy, Cuneo thinks, one successfully and fittingly (i.e., non-defectively) thanks God if one participates in a liturgical act of gratitude without the contrary attitude (i.e., resentment) towards God. This is because expressing gratitude is the *appropriate response* to what God has done on our behalf, regardless of how I feel about it. Thus, participating in the Eucharist without an attitude of resentment counts an instance of liturgical gratitude, so long as one is not participating whilst harbouring resentment against God in some capacity.

No doubt more needs to be said if we are to convince those who maintain that emotion is essential to any expression of gratitude that scripted liturgical gratitude is an appropriate response. In expanding Cuneo's notion of liturgical aptness/inaptness, it will be helpful to distinguish the grateful emotion from the grateful action. For as Gulliford et al. note, in "the earliest psychological writings about gratitude, there was no mention of a necessary emotional response. Bertocci and Millard defined gratitude as 'the willingness to recognize that one has been the beneficiary of someone's kindness, whether the emotional response is present or not.'" (Gulliford et al. 2013, p. 294; citing Bertocci and Millard 1963). While we are not advocating for a removal of talk about gratitude emotions, separating emotions from decisions can help us to examine more carefully how the two might be related.

Consider a distinction made in the psychological discussion of forgiveness. Everett L. Worthington, Jr. argues that "decisional forgiveness" (i.e., a wilful act to forgive someone who has wronged you) and "emotional forgiveness" (i.e., the feeling of no longer holding wrongdoing against a transgressor) can come apart in complex ways. Worthington writes, "People could decide to forgive and not experience emotional forgiveness. They also could experience sudden compassion for a transgressor . . . and realize that unforgiveness had

disappeared even though no decision had been made to forgive" ([Worthington 2013](#), p. 25). It is not always the case, Worthington thinks, that emotional forgiveness always precedes decisional forgiveness or vice versa. While it is typically the case that decisional forgiveness leads to emotional forgiveness, this might not always be so; one's affective response to a perpetrator might change and lead to a decision to forgive. Worthington lays out the similarities between the two kinds of forgiveness as such:

Decisional Forgiveness

(a)    Arrived at rationally or by will
(b)    May come before or after emotional forgiveness
(c)    May occur without emotional forgiveness
(d)    Aimed at controlling future behavior (not motives or emotions)
(e)    May make person feel "settled," calming emotion and motivation (i.e., might lead to emotional forgiveness or at least reduce emotional unforgiveness)
(f)    May give new meaning to situation
(g)    Changes behavior
(h)    May improve interactions by de-escalating or promoting reconciliation

Emotional Forgiveness

(a)    Arrived at by emotional replacement
(b)    Necessarily reduces unforgiving emotions
(c)    May come before or after decisional forgiveness (but usually after)
(d)    May occur without decisional forgiveness on rare occasions
(e)    Aimed at changing emotional climate but inevitably triggers neoassociationistic networks leading to changes in motives, thoughts, and other associations
(f)    May give new meaning to situation
(g)    May change behavior
(h)    Will change motivation
(i)    Makes person feel less negative emotionally and perhaps more positive
(j)    May improve interactions and promote reconciliation
(k)    May reduce the injustice gap
(l)    May reduce the justice motive
        ([Worthington 2013](#), Table 2.1, p. 59)

It seems plausible to think that a similar distinction might be made between decisional and emotional gratitude. We might apply these to claims to gratitude as follows:

Decisional Gratitude

(a)    Arrived at rationally or by will
(b)    May come before or after emotional gratitude
(c)    May occur without emotional gratitude
(d)    Aimed at controlling future behavior (not motives or emotions)
(e)    May make person feel "settled," calming emotion and motivation (i.e., might lead to emotional gratitude or at least reduce emotional ingratitude)
(f)    May give new meaning to situation
(g)    Changes behavior

Emotional Gratitude

(a)    Arrived at by emotional replacement
(b)    Necessarily reduces ungrateful emotions
(c)    May come before or after decisional gratitude (but usually after)
(d)    May occur without decisional gratitude on rare occasions
(e)    Aimed at changing emotional climate but inevitably triggers neoassociationistic networks leading to changes in motives, thoughts, and other associations
(f)    May give new meaning to situation
(g)    May change behavior
(h)    Will change motivation

(i)     Makes person feel less negative emotionally and perhaps more positive (Adapted from Worthington 2013, Table 2.1, p. 59)

Making this distinction allows us to say something more nuanced about the role of emotion in liturgical gratitude. It seems clear that liturgical gratitude always involves a kind of decisional gratitude, in which the participant chooses to engage in acts of thanksgiving to God. Moreover, in many cases of expressing liturgical gratitude it may be the case that participants do not respond emotionally to God. However, as the distinction above alludes to, participating in grateful liturgies by *deciding* to be grateful may still have the effect of increasing one's gratitude emotions in the long-term. As Worthington argues, acts of decisional forgiveness may *cause* emotional forgiveness; it seems reasonable to think something similar is going on in the case of gratitude. Indeed, as James KA Smith (2009) has argued extensively in his work on cultural liturgies, there is some value in participating in liturgy even if one is not "feeling it". Engaging in confession regularly, teaches one how to forgive and be forgiven. Petitioning God regularly teaches one how to pray. Similarly, liturgical acts of gratitude provide a kind of training for how to act gratefully to God. This may have the result of leading to an increase in emotional gratitude towards God, even if this outcome is not instantaneous. Indeed, one might argue that the function of a gratitude diary, a method used widely in the gratitude literature (see Emmons 2013, pp. 159–72 for an example), is precisely to develop gratitude in this way. By repeatedly *deciding* to record things that one is grateful for (regardless of one's feelings in that particular moment), one can become more grateful, in the sense of have grateful emotional experiences more often. It would be strange to suggest that gratitude is only really occurring much later down the line, only after one has been trained into emotionally responding to God in the appropriate way.

Thus, we think that it is important to not to dismiss liturgical gratitude as a bad instance of gratitude even if it lacks an emotional component. While it may be the case that the therapeutic effects of gratitude are lessened in these cases, there is still value in learning *how* to express gratitude in a way that is appropriate to the context. Similarly, encouraging young children to write thank you letters to their relatives after receiving birthday gifts each year is a good thing to do, even if they do so begrudgingly. The hope of the parents is that this practice of expressing gratitude becomes second nature to a child after years of learning how to respond; it is important for them to see that gratitude is the appropriate response to a gift. Moreover, it will also hopefully result in the development of emotional gratitude, especially if the child is encouraged to reflect on what it is they are doing when they write these letters.

Expressing gratitude in liturgy is just like this; we should do it even if we cannot always show-up emotionally, so to speak. A helpful example of this can be seen in the opening of the Church of England's Eucharistic liturgy:

Let us give thanks to the Lord our God

*All:* **It is right to give thanks and praise** (Church of England 2000)

The liturgy indicates that thanks is the *appropriate* response to God. It would be a high bar for participation if this meant that one must always *feel* grateful in one's response to God. Engaging in acts of thankfulness in liturgy is often an instance of decisional gratitude; deciding to thank God because it is right to do so. Liturgy encourages us to be grateful to God even when we do not feel grateful—even if we have had a terrible week, even if we are feeling doubtful and worried—we should be grateful because *it is right to give thanks and praise to God*. We must remain open to the fact that for many (as in Dunnington's discussion), emotional gratitude may never arise through liturgical gratitude. Engaging in liturgical gratitude will ideally lead to emotional gratitude to God, but it cannot be a requirement that it must always do so. We decide to express gratitude to God because it is right to do so.

In recognising the scripted nature of all liturgies, we have argued that liturgy provides a training ground for learning how to be grateful to God, even if one is not always feeling

grateful. Cuneo's account of liturgical action provides a helpful starting point for thinking about how acts of speaking, kissing, eating, and so on, can count as instances of thanking God liturgically. It is reasonable to assume, we think, that many of the therapeutic benefits of gratitude will follow from liturgical gratitude, even if these are formed over a long period of time.

### 3. Liturgical Gratitude Is a Species of Group Gratitude

The second distinctive feature of liturgical gratitude is that it is always a kind of group gratitude. Christian worship is always situated in the context of the community of the Church. As the theologian Evelyn Underhill describes, it is a central pillar of Christian thinking about worship to see that worship is never merely about individuals. She writes,

> The worshipping life of the Christian whilst profoundly personal, is essentially that of a person who is also a member of a group . . . The Christian as such cannot fulfil his spiritual obligations in solitude. He forms part of a social and spiritual complex with a new relation to God; an organism which is quickened and united by that Spirit of supernatural charity which sanctifies the human race from above, and is required to incarnate something of this supernatural charity in the visible world. Therefore even his most lonely contemplations are not merely private matter; but always to be regarded in their relation to the purpose and action of God Who incites them, and to the total life of the Church. (Underhill 1936, p. 83)

As the Apostle Paul stresses in his First Epistle to the Corinthians, "the body does not consist of one member but of many . . . If all were a single member, where would the body be? As it is, there are many parts, yet one body". (1 Cor 12: 11, 19). The context of Paul's discussion here is that of spiritual gifts given to the members of the Church; some are given gifts of healing, others, gifts of prophecy, and some the gifts of speaking in tongues. There are many different gifts, but the context in which these gifts are given is in the community of the Church. To make sense of the individual responses to God, we need to situate them in the context of the Church as the community of God. As Paul puts it two chapters earlier, we see this unity in the Church starkly in the celebration of the Eucharist: "we who are many are one body, for we all partake of the one bread" (1 Cor 10: 17).

This communal emphasis on the Church as the context of worship importantly shapes our understanding of liturgy. In Wolterstorff's words, "The church blesses God, praises God, thanks God, confess her sins to God, petitions God, listens to God's Word, celebrates the Eucharist. It's not the individual members who do these things simultaneously; it's the assembled body that does these things" (Wolterstorff 2015, p. 11). Thus, unlike many of the acts of practices recommended in the gratitude literature—gratitude journaling, spiritual disciplines, letter writing, etc.—liturgical gratitude takes place in the context of a group. As Tsang (2021) notes, those writing in field of gratitude have rarely paid attention to social contexts of gratitude. Thus, care is needed in thinking through this important feature of liturgical gratitude.

It will be helpful to begin with some examples:

(a) In the Church of England's liturgy for Holy Communion, the congregation are invited to say the following prayer together after receiving bread and wine:

> Almighty God,
>
> we thank you for feeding us
>
> with the body and blood of your Son Jesus Christ.
>
> Through him we offer you our souls and bodies
>
> to be a living sacrifice.
>
> Send us out
>
> in the power of your Spirit
>
> to live and work
>
> to your praise and glory.

Amen. (Church of England 2000)

(b) The first verse of the hymn written by American country artist, Jim Reeves begins as follows:

We thank Thee each morning for a newborn day

Where we may work the fields of new mown hay

We thank Thee for the sunshine

And the air that we breathe

Oh Lord we thank Thee (Reeves 1962)

(c) In a service in a low-church tradition, after the sermon on 1 Thessalonians 5:18 ("give thanks in all circumstances; for this is God's will for you in Christ Jesus"), the pastor stands up and prays using the following words:

Lord we are so grateful for what you are doing in the life of our community. Thank you for your Word and its challenge to us this morning to thank you in all circumstances. We pray you would make us a thankful people. In Jesus' name we pray, Amen.

How should we understand the meaning of these plural expressions of gratitude? In a recent article on group gratitude, we (Cockayne and Salter, forthcoming) offer a taxonomy for thinking about three different kinds of group gratitude. We think these distinctions can help expand our concept of liturgical gratitude. We offer the following distinctions:

Group-context gratitude: "gratitude is experienced or expressed in a group setting but the grateful agent is an individual, rather than a group"., e.g., imagine a "community stricken by natural disaster receiving money to fund the rebuilding of the whole town" in which an individual is grateful for the money received by the town. Here, "All that the group provides is the context for such gratitude". (Cockayne and Salter, forthcoming, p. 20)

Joint gratitude: "Joint gratitude involves 1) jointly attending to the source of gratitude, 2) co-attenders actively signalling their grateful attitude to that source (even if this attitude is not always identical in all participants) and 3) jointly responding with some kind of grateful action". E.g., "If . . . a couple received a gift from a friend, they could express gratitude individually; one might bake the generous friend a cake, and the other could write the friend a letter". (Cockayne and Salter, forthcoming, pp. 22–23)

Collective gratitude: "Collective gratitude occurs when organisations or social groups are organised such that they can act gratefully in response to benefits identified at the collective level. Note that unlike the other kinds of group gratitude in the taxonomy, collective gratitude is not dependent on joint attention, and we assume that there is no collective-level phenomenology. The individuals on whom the collective actions depend on may attend to group-level benefits, but the collective as the subject of gratitude can only *identify* benefits through decision-making procedure (such as voting, or group hierarchy)". e.g., "Suppose a university receives a large financial gift from one of its donors. The university hierarchy meets together to decide what the best response to this donation might be, and after some deliberation, decide to send a letter on behalf of the university, as well as naming one of their faculty builds after the benefactor. In the letter, the University Principal writes the following words: "On behalf of the university I would like to express my deep gratitude for your donation; this gift will benefit many students for many years to come"". (Cockayne and Salter, forthcoming, pp. 16, 26)

As we argue, "many cases will be difficult to map neatly onto this taxonomy. For instance, in cases of collective action it may be that all three kinds of group gratitude are occurring simultaneously—an organisation which displays collective-level gratitude may

do so through the joint actions of many individuals who are also experiencing gratitude individually" (Cockayne and Salter, forthcoming, p. 28). Nevertheless, this taxonomy seems useful in understanding the different ways in which group gratitude may be present in expressions of liturgical gratitude.

First, many of the examples discussed in the first section of this paper (such as expressing thanks by participating in the Eucharist in the appropriate manner) would seem to be instances of what we call "group-context gratitude". Cuneo's analysis focuses predominantly on how individuals might express liturgical gratitude in the context of corporate worship, but the focus is primarily on the individual expressing gratitude in the liturgy. This is the weakest form of group gratitude (in the sense that all the group contributes is the context for expressing gratitude), but clearly provides one way of making sense of the corporate claims made about liturgy by Underhill and others: gratitude is always expressed in the context of a group for one's relationship to God is always bound up in one's relationship to the Church as Christ's body. In many cases of liturgical gratitude, it may be sufficient simply to point to the context in which individuals are expressing thanks to God.[4] Yet, this notion of group-context gratitude is not the only kind of group gratitude occurring in liturgical contexts.

Secondly, then, there also seem to be many cases of joint gratitude in liturgy. Indeed, Cuneo argues that in the example of singing in liturgy, we are required to sing *together* not merely to sing as individuals; "to engage in group singing . . . requires that I adjust my singing to yours and that you adjust your singing to mine in 'real time', often in ways that are not dictated by the score that we are following". (Cuneo 2016, p. 138). In liturgical singing there are joint intentions present, such that individuals intend to sing the liturgy *together*, rather than intending to act as individuals. The difference is subtle, but significant. Consider an example from John Searle to why this is the case:

> Imagine that a group of people are sitting on the grass in various places in a park. Imagine that it suddenly starts to rain and they all get up and run to a common, centrally located, shelter. Each person has the intention expressed by the sentence "I am running to the shelter". But for each person, we may suppose that his or her intention is entirely independent of the intentions and behavior of others. In this case, there is no collective behavior; there is just a sequence of individual acts that happen to converge on a common goal. Now imagine a case where a group of people in a park converge on a common point as a piece of collective behavior. Imagine that they are part of an outdoor ballet where the choreography calls for the entire *corps de ballet* to converge on a common point. We can even imagine that the external bodily movements are indistinguishable in the two cases; the people running for shelter make the same types of bodily movements as the ballet dancers. Externally observed the two cases are indistinguishable, but they are clearly different internally. (Searle 1990, pp. 403–4)

While in cases of group-context gratitude, individuals may happen to express the same intention at the same time, in acts of joint liturgical gratitude, individuals attempt to mesh or merge their intentions, such that the subject expressing gratitude is "we", not "I". If I try to speak at the same time as my fellow congregants, or to co-ordinate my movements such that I raise my hand in worship when the worship leader sings the line: "so we raise up holy hands to praise the holy One", then liturgical gratitude is a kind of joint gratitude.

Note that on an account of joint liturgical gratitude, it can sometimes be difficult to pick out precisely who is included in the pronoun, "we". For it might be the case that the "we" in liturgical contexts extends to anyone included in my joint intention. For example, in singing a hymn of gratitude, I might intend to sing with the whole congregation; here the meaning of "we" seems to extend to all those contained in my intention. However, if I am a member of the choir seated at a distance from the rest of the congregation whom I am not consciously attending to, then plausibly, the meaning of "we" might only extend to the members of the choir. In other words, there are limits to joint gratitude. These analyses fail to capture all instances of group gratitude in liturgy.

The problem is not unique to liturgy either; as Stephanie Collins (2019) describes, joint action "rises and falls with the specific joint commitment that defines it—for example, a joint commitment to paint the house or go for a walk" (p. 56). Joint action analyses are helpful at explaining cases of actions performed by small groups, such as moving furniture or performing pieces of music, but they stop short of providing explanations for when group behavior is dispersed, or in which the "we" described persists beyond the specific actions performed. For example, a joint action account will fare poorly in explaining how it is that an organization like a university can express thanks to all its staff.

Thirdly, then, it seems that the liturgical "we" refers sometimes to a broader group than is captured by the joint-action account. We think that there may also be cases of collective gratitude present in liturgical contexts. At times, the subject of liturgical expressions of gratitude does not appear to refer to either those jointly enacting the liturgy or individuals who are enacting the liturgy in a group context. Consider the prayer we gave in example (c):

> Lord we are so grateful for what you are doing in the life of our community. Thank you for your Word and its challenge to us this morning to thank you in all circumstances. We pray you would make us a thankful people. In Jesus' name. Amen.

The expression of gratitude in this prayer seems to refer not only to those jointly enacting the liturgy, but also to the community more widely. For instance, Betty, a long-standing member of the congregation, might have been in bed with a fever on the morning of this prayer, but arguably the pastor's prayer still applies to her as she is a member of the community. Similarly, Jon may have been distracted when the prayer was being said and fail to attend to the pastor's words. But the words still apply to him. In other words, group gratitude sometimes describes an action that is performed by a wider community, beyond instances of joint action. These actions might lead to groups developing a culture of gratitude. To specify what this culture amounts to, we need to say something about how a group is capable of acting. Consider an example from our previous work:

> Consider how a newspaper might be said to display the virtue of courageous journalism. While a team of investigative journalists might display courage through joint actions (such as the exposure of abuse by the *Boston Globe* as depicted in the movie, *Spotlight*), we might also say of a newspaper that it is courageous over a long period of time (i.e., we might say that the *Boston Globe* has consistently acted courageously in pursuit of the truth for the past two decades, even though its editorial team have changed entirely over this period). Joint action accounts do not allow us to say much of the long-term actions or virtues of a group if there are changes in constitution. (Cockayne and Salter, forthcoming, p. 15)

To make sense of such claims, we argue, we can think of groups as collectives, that is, agents capable of acting together through their organisational structure. For example, a government decides to enact certain policy by voting on the best course of action, or by deferring to an individual with ultimate authority, like a Prime Minister or President. In collectives, there are typically those who act on behalf of the group, and those who authorize others to act on their behalf. For example, as Christian List and Philip Pettit argue,

> In a participatory group like a voluntary association, members have the same status within the group agent; they equally authorize the group agent and take roughly equal parts in acting on its behalf. In a hierarchical organization, such as a commercial corporation or a church, there may be differences in the members' roles, for example through holding different offices or through belonging to subgroups with different tasks. (List and Pettit 2011, p. 36)

How does this help us to think about liturgical gratitude? Consider how churches make decisions as collectives. Typically, most churches have decision-making bodies (church councils, eldership bodies) which vote on which actions the church is to perform, but they may also authorize individuals to make decisions on behalf of all. For example, a worship leader may be authorized to choose the hymns for Sunday services. In these decisions,

whether they are made through voting or authorization, there are choices made about whether to act gratefully. What balance should be struck between praise and confession in the music used in worship? Should there be space in liturgy to hear testimonies of individual thanks? Should the Church express thanks to members in the community as a group through public statements and letters of thanks? These decisions will all impact the grateful actions of the church as a collective, but also the culture of the community itself.

Finally, this broader sense of "we" in the context of liturgical gratitude may extend to a much broader group, namely, the body of Christ. Return to Underhill's remarks that in worship we, form "part of a social and spiritual complex with a new relation to God; an organism which is quickened and united" by the Holy Spirit (Underhill 1936, p. 83). In expressing gratitude to God in liturgy, we are brought into a group that extends beyond what we are currently aware of either in congregational structure, or through jointly intentional action.

Thus, we have seen that liturgical gratitude is complex. There are many different ways of trying to understand the group dynamics present in liturgical gratitude. Future work on liturgical gratitude (empirical or not) will need to be sensitive to these distinctions; rather than thinking of group gratitude as a monolithic concept, it is important to see how the context of the gratitude expression makes a difference to the group dynamics present, as well as the ways in which the individual expresses gratitude in this context.

## 4. What Are the Effects of Liturgical Gratitude?

We have now considered how to make sense of liturgical gratitude by considering two distinctive features of liturgical gratitude. First, its scriptedness, and second, the fact that liturgical gratitude is always a kind of group gratitude. Clearly, there will be other instances of gratitude which are either, or perhaps both, scripted and social. However, unlike many of the cases of individual gratitude discussed in the psychological literature, liturgical gratitude always has these distinctive features present. This raises the possibility that there are important empirical effects of liturgical gratitude on those who participate, which are not always present in individual gratitude practices. We conclude by offering three ways in which liturgical gratitude might provide an effect on its participants. We hope these will provide impetus for future empirical studies of liturgical gratitude as a distinct context for gratitude behavior.

The first claim is that individuals are more likely to experience strong feelings of gratitude to God when experiencing these feelings in a group context. Studies have found that both positively- and negatively-valanced experiences are felt more intensely when shared with a partner. Garriy Shteynberg and colleagues (Shteynberg et al. 2014) found that sharing scary, sad and happy stimuli with another induced stronger emotional experiences of each kind. A similar pattern can be observed with sensory experiences; individuals rate chocolate as more likeable and flavorful having tasted it simultaneously with another, and rated an unpleasantly bitter chocolate as less likeable having tasted with another (Boothby et al. 2014). These studies provide evidence that shared experiences are not simply more enjoyable than solitary experiences but intensify emotional and sensory experiences.

It is reasonable to extend this logic to gratitude; grateful feelings are amplified when shared. In the context of grateful liturgical practices, it is plausible that this amplification will apply in cases of joint gratitude. That is, rather than a situation in which individuals in a group setting are encouraged to bring to mind reasons they are each individually grateful to God, all members of the group express gratitude to God together. This is not to claim that shared grateful experiences are *always* stronger than individual grateful experiences, but that sharedness is just one of many factors that might influence an individual's emotional experience.

The second claim, closely related to the first, is that expressing gratitude to God through liturgy as part of a group leads to greater positive effects on social bonding than those observed in cases of individual gratitude. The sharing of emotional experiences creates social closeness (Zahavi and Rochat 2015), as does shared activity of various kinds,

from synchronous bodily movements to planned joint actions (McNeill 1997; Hove and Risen 2009; Whitehouse and Lanman 2014). However, it is possible to go further than the claim that shared gratitude promotes social affiliation because shared emotions and actions in general promote social affiliation. Gratitude arguably has specific influences on social cohesion. While it has been highlighted how gratitude serves to promote social affiliation between benefactor and recipient, it is also plausible that there are effects that promote cohesion within larger groups (Algoe 2012; Algoe et al. 2008; Tsang 2021). For example, individuals that experience gratitude towards a fellow group member (such as a fellow sorority member) also feel more integrated within that social group (the sorority) (Algoe et al. 2008). A proposed reason for this is that a dyadic-level sense of social value that comes with gratitude serves to enhance one's sense that they are a genuine member of the group. It is plausible that this is also the case at the joint level; sharing gratitude with others helps create a sense of being part of an integrated group that experiences common benefits.

Here, particular features of gratitude to God are also relevant. Tsang (2021) suggests that group-based benefits (those that help an individual but are not specifically for that individual, such as healthcare, civil liberties, etc.) typically generate weaker grateful feelings than individual benefits (e.g., a benefactor paying for another's operation). She suggests that this because in the individual case it is clearer how a specific relationship benefits, which has the further effect of there being a clearer sense of a benevolent motivation. However, as Tsang goes on to suggest, if the benefactor is God, it is possible that a specific sense of relationship, and thus benevolence, can emerge. We might also expect different effects in the case of liturgy, where the sense of group is both at the collective level and at the joint level. Here, the sense of the group in which one is benefitting may be collective (i.e., the Church), but also distinctly local (i.e., this church), making the sense of group benefit much more specific. Thus, cases of *joint* liturgical gratitude have a specific benefactor (God) and a specific recipient (this community), which may help create a clearer and thus stronger sense of gratitude amongst that group, which in turn facilitates stronger integration amongst members of that community.

A further step is to follow Emmons and Stern's (2013) distinction between gratitude's "worldly" and "transcendent" definitions. Focusing on the latter, Emmons and Stern quote Streng (1989), who states that those who adopt a grateful attitude " . . . recognize that they are connected to each other in a mysterious and miraculous way" (Streng 1989, p. 5). Emmons and Stern thus define transcendent gratitude as " . . . the feeling of connection with humanity emerging from a sense of wonder and joy that participating in an intricate network of existence brings". (Emmons and Stern 2013, p. 847). If gratitude in its transcendent sense is a spiritual attitude pertaining to a sense of one's connectedness with others, it is plausible that this sense of social connectedness is magnified when experiencing a gratitude with others. The presence of others is an especially tangible reminder of one's connectedness, especially when those others are engaging in a shared grateful practice with. This may also be particularly so in the case of group gratitude to God, given that in such cases the transcendent sense of gratitude is salient even if the source of gratitude is something mundane (e.g., "thank you that our leaky roof has been fixed").

The third and final claim is that participating in acts of group liturgical gratitude can serve as a means of facilitating gratitude in individuals who are not experiencing gratitude to God as an individual. As previously highlighted, the gratitude literature already assumes that by engaging in intentional gratitude practices (such as journaling; Emmons 2013), an individual might facilitate an increase in grateful feelings, regardless of their emotional state at the point of deciding to engage in such practices, or engaging directly in the practice. We propose a similar process might occur in the case of participating in liturgy that expresses thanks. An individual participant who is not feeling grateful during the process of engaging in the grateful liturgy may nonetheless experience an increase in grateful feelings if they continue to participate in the practice over time. However, beyond the effectiveness of the practice itself, it is plausible that embedding that non-grateful individual within a group of grateful others might be a further avenue for increased gratitude. Participating in a shared

expression of gratitude like a liturgical practice may serve to align the attitude of the non-grateful participant with the attitude of the grateful participants around them, a process which has been argued to be an effect of participating in joint actions (Gallotti et al. 2017).

**5. Conclusions**

Given that liturgy is one of the primary places in which gratitude is expressed in the life of the Christian, it seems pressing to understand the nature of liturgical gratitude. We have argued that there are two features of liturgical gratitude that distinguish it from many typical instances of gratitude discussed in the existing literature. Firstly, liturgical gratitude is scripted, which leaves open the possibility that one might express gratitude even if they are not feeling grateful emotions. Here, we wish to resist the typical view that gratitude must always be accompanied by a corresponding emotional state; deciding to act gratefully in liturgy is the appropriate response to God because God deserves our gratitude, even if we are not feeling so grateful. Moreover, in deciding to be grateful, we argued, we create a context in which we can provide further opportunities to feel grateful, much like training a child how to respond appropriately to the social convention of gift giving. Secondly, liturgical gratitude always takes place in the context of a group. If we recognize the theological importance of seeing all worship as part of the community of the Church, then we should also see that liturgical gratitude is a kind of group gratitude. We have shown that group gratitude is a complex phenomenon (or set of phenomena) and that paying attention to the context is crucial for understanding the meaning of expressions of liturgical gratitude. Finally, we have argued that identifying these two distinctive features means that there are some important features to liturgical gratitude which we expect to result in distinctive empirical effects.

**Author Contributions:** Writing—original draft preparation, J.C. and G.S.; writing—review and editing, J.C. and G.S.; funding acquisition, J.C. All authors have read and agreed to the published version of the manuscript.

**Funding:** This research This research was generously funded by the John Templeton Foundation, Grant No. 61513.

**Conflicts of Interest:** The authors declare no conflict of interest.

**Notes**

1　Arguably, as one anonymous reviewer helpfully highlights, such concerns regarding sincerity are primarily (or indeed only) an issue if assuming a Protestant view of Christian worship, whereby extemporaneous expressions of worship are often viewed as more sincere than formal, composed texts. Regardless, it is still helpful to our account to articulate a positive account of liturgical expressions of gratitude as genuine, both for addressing potential theological concerns (from a Protestant perspective or otherwise) and for addressing the potential concerns of psychologists who place a strong emphasis on gratitude requiring an emotional component (e.g., Emmons 2008).

2　Indeed, Kent Dunnington (forthcoming) has recently argued that gratitude to God does not require us to have emotional gratitude states for precisely this reason.

3　There are some who take a more general account of liturgy (see Smith 2009) to refer to any goal-oriented ritual. We refer to liturgy in its restricted sense in the context of gathered Christian worship, as in Wolterstorff's definition.

4　Note that group-context gratitude best captures the kinds of group gratitude considered in Tsang's (2021) study.

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
