# Peer review of "Liturgical Gratitude to God"

_religions, doi:10.3390/rel13090795_

Round 1

Reviewer 1 Report

The paper is interesting - the author(s) found a way to explain why liturgical gratitude in religions (especially in Christianity) could be explained by (1) scripts and (2) social influence. While the article was not supported with empirical data, the review of the topic was succinctly written. The author(s) offer several reviews (e.g., book chapter) to support the argument. If the author(s) could include empirical supports would help to strengthen the paper even further. For example, the comparisons between forgiveness and gratitude in the domain of emotional and decisional approach could be enhanced with empirical data. Despite that, the paper has demonstrated their argument well.

Author Response

We have reviewed the paper in line with the referee's comments. 

Reviewer 2 Report

The article offers an interesting perspective on gratitude. There are several places where it would benefit from some editing, and the whole from some careful copy editing. 

Some specific questions/suggestions:

1. I found the "always scripted" claim somewhat odd, but it makes sense after you finally define what you intend by "scripted." That definition needs to occur earlier in the argument, and perhaps put into conversation with recent definitions of "ritual" and "ritual action". 

2. Your brief discussion of "sincerity" suggests a very Protestant reading of Christian worship, e.g., that formal composed texts lack sincerity and that extemporaneous expressions are to be preferred. You address this implicitly, but could perhaps take it on more directly.

3. Your point about eucharistic action (eating/drinking) as thanksgiving seems to miss the point of the eucharistic prayer as a prayer of thanksgiving.

4. Line 97, quote from Wolterstorff "the count-as significance" phrase doesn't make sense. Please check and clarify the quote.

5. The discussion of "we" in lines 390-425 seems a tangent; you return to the point later (line 510) with much more clarity and a more apt discussion. Nothing is lost from the argument by deleting these lines.

6. Line 180: most commentators today would argue that the question of worthiness in 1 Cor. has nothing to do with an attitude of thanksgiving but of awareness of the body of Christ in the gathered community.

Author Response

  1. I found the "always scripted" claim somewhat odd, but it makes sense after you finally define what you intend by "scripted." That definition needs to occur earlier in the argument, and perhaps put into conversation with recent definitions of "ritual" and "ritual action". 

We have clarified this in the introduction and throughout

2. Your brief discussion of "sincerity" suggests a very Protestant reading of Christian worship, e.g., that formal composed texts lack sincerity and that extemporaneous expressions are to be preferred. You address this implicitly, but could perhaps take it on more directly.

We have added a footnote addressing this point

3. Your point about eucharistic action (eating/drinking) as thanksgiving seems to miss the point of the eucharistic prayer as a prayer of thanksgiving.

We have added a clarification to acknowledge this point

4. Line 97, quote from Wolterstorff "the count-as significance" phrase doesn't make sense. Please check and clarify the quote.

This has been clarified

5. The discussion of "we" in lines 390-425 seems a tangent; you return to the point later (line 510) with much more clarity and a more apt discussion. Nothing is lost from the argument by deleting these lines.

These lines have been removed

6. Line 180: most commentators today would argue that the question of worthiness in 1 Cor. has nothing to do with an attitude of thanksgiving but of awareness of the body of Christ in the gathered community.

This discussion has been removed